# Low-Grade Gliomas: Histological Subtypes, Molecular Mechanisms, and Treatment Strategies

**DOI:** 10.3390/brainsci13121700

**Published:** 2023-12-09

**Authors:** Corneliu Toader, Lucian Eva, Daniel Costea, Antonio Daniel Corlatescu, Razvan-Adrian Covache-Busuioc, Bogdan-Gabriel Bratu, Luca Andrei Glavan, Horia Petre Costin, Andrei Adrian Popa, Alexandru Vlad Ciurea

**Affiliations:** 1Department of Neurosurgery, “Carol Davila” University of Medicine and Pharmacy, 020021 Bucharest, Romania; corneliu.toader@umfcd.ro (C.T.); antonio.corlatescu0920@stud.umfcd.ro (A.D.C.); razvan-adrian.covache-busuioc0720@stud.umfcd.ro (R.-A.C.-B.); bogdan.bratu@stud.umfcd.ro (B.-G.B.); luca-andrei.glavan0720@stud.umfcd.ro (L.A.G.); horia-petre.costin0720@stud.umfcd.ro (H.P.C.); andreiadrianpopa@stud.umfcd.ro (A.A.P.); prof.avciurea@gmail.com (A.V.C.); 2Department of Vascular Neurosurgery, National Institute of Neurology and Neurovascular Diseases, 077160 Bucharest, Romania; 3Department of Neurosurgery, Dunarea de Jos University, 800010 Galati, Romania; 4Department of Neurosurgery, Clinical Emergency Hospital “Prof. Dr. Nicolae Oblu”, 700309 Iasi, Romania; 5Department of Neurosurgery, “Victor Babes” University of Medicine and Pharmacy, 300041 Timisoara, Romania; 6Neurosurgery Department, Sanador Clinical Hospital, 010991 Bucharest, Romania

**Keywords:** low-grade gliomas, astrocytoma, oligodendroglioma, ependymoma, rare low-grade gliomas, pediatric low-grade gliomas, neuropathology and classification, molecular pathways, outcome, treatment strategies, surgery, radiation therapy, targeted therapies, immune therapies

## Abstract

Low-Grade Gliomas (LGGs) represent a diverse group of brain tumors originating from glial cells, characterized by their unique histopathological and molecular features. This article offers a comprehensive exploration of LGGs, shedding light on their subtypes, histological and molecular aspects. By delving into the World Health Organization’s grading system, 5th edition, various specificities were added due to an in-depth understanding of emerging laboratory techniques, especially genomic analysis. Moreover, treatment modalities are extensively discussed. The degree of surgical resection should always be considered according to postoperative quality of life and cognitive status. Adjuvant therapies focused on chemotherapy and radiotherapy depend on tumor grading and invasiveness. In the current literature, emerging targeted molecular therapies are well discussed due to their succinctly therapeutic effect; in our article, those therapies are summarized based on posttreatment results and possible adverse effects. This review serves as a valuable resource for clinicians, researchers, and medical professionals aiming to deepen their knowledge on LGGs and enhance patient care.

## 1. Introduction

The prognosis for patients with lower-grade diffuse gliomas (LrGGs), classified as grades II and III, is showing signs of improvement, though it varies based on the molecular subtype of the tumor. Despite these advancements in survival, both the tumors themselves and the treatments employed to combat them frequently result in considerable cognitive impairments. These impairments can be both objective (measurable through cognitive testing) and subjective (as they are perceived by the patients themselves). Neoplasms of the central nervous system (CNS) are categorized according to their cellular origin and distinct histological characteristics, which are indicative of their probable clinical course. Among these neoplasms, gliomas, which arise from CNS glial cells, constitute a significant subgroup. These glial neoplasms are further divided into astrocytomas, oligodendrogliomas, mixed oligo-astrocytic, and mixed glioneuronal tumors, with each originating from different glial cell types, such as astrocytes or oligodendrocytes. The World Health Organization (WHO) employs a grading system for gliomas that spanns from grade 1 (least aggressive) to grade 4 (most aggressive) and is based on a range of histological characteristics, including cellular atypia, proliferative patterns, and necrosis presence. Specifically, low-grade gliomas (LGGs) are classified as grade 1 gliomas, which are devoid of these histological markers, or grade 2 gliomas, which exhibit only cellular atypia [1].

Low-grade astrocytic tumors include diffuse astrocytomas, pilomyxoid astrocytomas, and pleomorphic xanthoastrocytomas (WHO grade 2), as well as SEGA and pilocytic astrocytomas (WHO grade 1). Oligodendrogliomas and oligoastrocytomas (WHO grade 2) represent low-grade oligodendroglial tumors. Additionally, specific low-grade glioneuronal tumors such as gangliogliomas and dysembryoplastic neuroepithelial tumors are categorized under WHO grade 1 [2]. In the revised taxonomy of diffuse gliomas, a substantial proportion have been reclassified based on IDH 1/2 mutation status and the 1p/19q codeletion, leading to the anticipated redundancy of the oligoastrocytoma category and the redefinition of gliomatosis cerebri as a growth pattern [3].

Assessing the incidence of low-grade gliomas poses a challenge due to the recent transition to a molecular-based classification. Cancer registries are gradually integrating changes from the 2016 WHO neuropathological categorization. Based on previous classifications, the estimated yearly incidence rates in the U.S. for grade 2 astrocytomas, oligodendrogliomas, and mixed gliomas are 0.51, 0.25, and 0.20 per 100,000 individuals, amounting to 1180, 690, and 610 cases, respectively. There is a higher prevalence of low-grade gliomas among white people compared to people, with lower rates in American Indians/Alaska Natives and Asian/Pacific Islanders. Astrocytomas commonly peak between ages 30 and 40 years old, whereas oligodendrogliomas peak at ages 40–45. Males are slightly more affected by low-grade gliomas [4,5].

The precise etiological factors for low-grade gliomas are not fully understood. Exposure to ionizing radiation, particularly among childhood leukemia survivors, is a recognized environmental risk factor. Intriguingly, a history of allergies or asthma seems to confer some protective effect against gliomas, suggesting the potential involvement of the immune system. Although rare inherited tumor syndromes contribute to a minority of cases, familial glioma occurrences and research pointing to increased glioma risk in close relatives imply more complex genetic factors. Recent genome-wide associations have identified gene variants correlated with a heightened risk of gliomas, including low-grade types. Notably, the g allele of CCDC26 on chromosome 8 elevates the risk of specific gliomas sixfold. This allele is present in about 40% of patients with certain glioma types compared to 8% in the general population. The mechanism of this variant is yet to be elucidated, and due to the overall low incidence of glioma, screening for this allele is not currently recommended [6].

## 2. Historical Overview of the 2021 WHO Classification: Molecular Intricacies and the Pathway to Targeted Therapies

Molecular advancements have substantially addressed the complexities in brain tumor classification. As a result, many brain tumors are now characterized by distinct molecular alterations. The 2021 5th Edition of the WHO Classification of Tumors of the Central Nervous System enhances the fundamental shifts introduced in the 2016 4th Edition, recognizing several new tumor entities, each assigned an official WHO grade. This edition notably incorporates methylome profiling, particularly pertinent for low-grade gliomas and glioneuronal tumors. The 2021 classification is significant for several reasons, such as its integration of molecular markers, improved diagnostic accuracy, the potential for personalized treatment, advancements in research, and potential better patient outcomes. These neoplasms are frequently categorized based on specific genetic changes such as FGFR1, MYB/MYBL1, BRAF, or IDH1/2, identified through DNA methylation profiles [7]. For pediatric-type low-grade gliomas and glioneuronal tumors (pLGG/GNTs), evidence suggests that MAP kinase pathway alterations are prevalent, albeit with variable manifestations and not always definitively. The 2021 WHO classification reflects the advancing comprehension of these tumors, where a specific genetic alteration can define a tumor, aid in its diagnosis, be common across different tumors, or be one of several alterations within a tumor type. Some diagnoses may not necessitate any demonstrated alteration, while others are yet to be discovered. The 2021 WHO’s “hybrid taxonomy” encapsulates the current understanding of CNS tumors’ clinical, histological, and molecular aspects, paving the way for more precise tumor classification and targeted therapies. The classification organizes gliomas, glioneuronal tumors, and neuronal tumors into six families. Three of these families correspond with pLGG/LGNT: pediatric-type diffuse low-grade gliomas [5], circumscribed astrocytic gliomas [8], and glioneuronal and neuronal tumors [9]. Moreover, six of the fourteen newly recognized tumor types in the 2021 WHO classification are categorized as pLGG/GNTs. Under “pediatric type diffuse low-grade gliomas,” three new tumor types are introduced: “diffuse astrocytoma, MYB or MYBL1-altered”; “polymorphous low-grade neuroepithelial tumor of the young (PLNTY)”; and “diffuse low-grade glioma-MAPK altered”. The category of glioneuronal and neuronal tumors includes three new additions: “Diffuse glioneuronal tumor with oligodendroglioma-like features and nuclear clusters (DGONC)”; “myxoid glioneuronal tumor (MGT)”; and “multinodular and vacuolating tumor (MVNT)” [10].

Patients with low-grade gliomas typically present at a younger median age compared to those with anaplastic gliomas or glioblastomas, usually diagnosed in their late twenties to mid-forties, although diagnosis over the age of 60 is possible. Seizures, ranging from generalized tonic–clonic to subtle partial seizures, are a frequent symptom, particularly in cases with oligodendroglial histology, likely due to their frequent cortical involvement. The widespread availability of CT and MRI scans has led to the incidental diagnosis of many patients while seeking care for unrelated conditions like migraines or head injuries [11,12]. Low-grade gliomas rarely present with specific focal deficits such as speech difficulties or unilateral weakness, as these tumors tend to infiltrate rather than disrupt critical brain structures. Neuroimaging is typically indicative of a low-grade glioma. Over 95% of these tumors are located in the cerebral hemispheres, with a near-even distribution across the frontal and temporal lobes and fewer in the occipital lobe. In CT scans, these tumors often appear as hypodense areas. Approximately 20% of low-grade gliomas, especially oligodendrogliomas, demonstrate calcification on CT. Additionally, around a quarter of these tumors show some contrast enhancement on CT, usually presenting as patchy rather than ring-like enhancement. Originating primarily in the white matter, those with an oligodendroglial component may extend into the cortex. MRI is more effective than CT in delineating these tumors, typically appearing as T1-hypointense and T2/FLAIR-hyperintense. MRI’s susceptibility-weighted imaging can detect calcifications or occasional hemorrhages which are more common in oligodendrogliomas than astrocytomas. Advanced imaging techniques like PET scanning and magnetic resonance spectroscopy can aid in differentiating tumor types, though they are not always necessary. A key subject of ongoing research is utilizing magnetic resonance spectroscopy to monitor low-grade glioma progression and response to treatment by identifying elevated levels of 2-hydroxyglutarate in IDH mutant gliomas [13].

Traditionally, diffuse infiltrating gliomas were identified and classified based on their morphological characteristics, which can be observed under light microscopy following hematoxylin and eosin (H&E) staining. Tumors characterized by elevated cellular density and nuclear atypia but with sparse mitotic figures were classified as low-grade gliomas. However, the subjective assessment of “rare” mitotic activity led to inconsistencies in grading by neuropathologists. To mitigate this, the Ki-67 stain, which marks proliferating cells, is utilized, with low-grade gliomas typically exhibiting less than 10% labeling. Further classification into subtypes like low-grade astrocytoma, oligodendroglioma, or oligoastrocytoma is achieved through the tumor’s cellular architecture and immunohistochemical staining [14]. Astrocytomas are noted for their pronounced fibrillary structures and strong reactivity to specific protein markers, whereas oligodendrogliomas possess scant cytoplasm and characteristic “fried egg” nuclei [15].

The advent of molecular neuropathology has profoundly augmented the understanding and categorization of low-grade gliomas. TP53 mutations, frequently observed in astrocytomas but uncommon in oligodendrogliomas, were among the early molecular distinctions recognized. The 1990s uncovered that most oligodendrogliomas exhibit distinct chromosomal losses, findings generally exclusive to TP53 mutations [16]. This led some experts to advocate for a molecular-based classification, positing it to be more objective and reflective of tumor behavior. A pivotal discovery was the prevalence of mutations in the IDH gene, involved in the Krebs cycle, in a majority of low-grade gliomas. The frequent IDH1 R132H mutation, in particular, can be readily detected, offering significant insights into glioma pathogenesis [17,18,19].

Within the WHO 2021 framework, histopathological grading adheres to the principles set by the WHO 2016 criteria, with the presence of necrosis and/or microvascular proliferation indicative of a grade 4 tumor, specifically classified as astrocytoma IDH mutant CNS WHO grade 4. Despite this continuity, a definitive criterion for differentiating grades 2 and 3 based on mitotic count remains unestablished [20]. Furthermore, while the Ki-67/MIB-1 proliferative index correlates with tumor grade, it lacks a universally accepted threshold for predicting increased recurrence risk [21]. In this context, the category of diffuse astrocytoma, IDH-wild-type, corresponding to CNS WHO grades II or III but lacking glioblastoma molecular characteristics, is now considered rare and has been removed from the CNS WHO5 classification [22]. Recent studies have led to the reclassification of IDH mutant grade 2 and 3 astrocytomas as “diffuse low-grade astrocytomas,” owing to their prognostic similarities. This reclassification questions the previous grouping of grade 3 and 4 astrocytomas as “high-grade,” given the distinct differences in molecular profiles and clinical outcomes between IDH mutant grade 3 astrocytomas and IDH-wild-type grade 4 glioblastomas [4].

Genomic analyses have shown that the majority of grade II and III diffuse astrocytomas, IDH-wild-type, harbor genomic alterations and clinical outcomes akin to primary glioblastoma, grade IV [23]. One particular study indicated that histopathologic grade II or III IDH-wild-type diffuse astrocytic gliomas, characterized by chromosomal anomalies such as +7/−10, EGFR amplification or TERT promoter mutations, are prognostically equivalent to histologically confirmed glioblastoma [24]. Moreover, the diagnosis of IDH mutant diffuse astrocytoma grade 2 is now strictly limited to cases without anaplastic histopathological features, significant mitotic activity, and the homozygous deletion of CDNK2A/B [25].

In-depth genomic investigations regarding various cancers have revealed a previously underappreciated prevalence of molecular alterations affecting the cellular epigenome [26]. This epigenome comprises DNA modifications, histones, their associated marks, and other chromatin-binding factors, all of which collectively orchestrate gene expression. The critical role of epigenomic dysfunction has been identified in several primary brain tumors, including gliomas [27]. Among these, mutations in isocitrate dehydrogenase 1 and 2 (IDH1 and IDH2) and the H3.3 histone-encoding genes H3F3A and HIST1H3B are particularly notable. IDH mutations result in a widespread pattern of DNA and histone hypermethylation, owing to the generation of the oncometabolite 2-hydroxyglutarate [28,29], while H3.3 mutations directly impact histone marks, chromatin accessibility, and gene expression [30]. These disruptions, complex and cell-specific, appear to fundamentally deviate from normal developmental pathways, contributing to the pathogenesis of glioma. Although pivotal in adult and/or high-grade glioma variants, IDH and H3.3 mutations are infrequently associated with pediatric low-grade gliomas (pLGG). Some studies have detected H3.3 K27M mutations in subgroups of pilocytic astrocytomas and glioneuronal tumors, which are typically more aggressive than their H3.3-mutant counterparts. Yet, these pLGG variants often have longer patient survival compared to high-grade gliomas with H3.3 mutations [31].

Epigenomic profiles have become indispensable markers in pLGG and other primary CNS tumors. Specifically, global DNA methylation profiling has enabled the identification of distinct “signatures” that often define brain tumor subtypes, laying the groundwork for systematic pLGG classification. Recent research employing global methylation profiling has been instrumental in characterizing various gliomas, and similar methodologies are expected to further refine pLGG classification in the future [32].

The discovery of key genetic alterations in pLGG has opened avenues for targeted therapies, particularly those addressing the commonly altered MAPK pathway in these tumors (Figure 1).

Selumetinib (AZD6244), an oral MEK1/2 inhibitor, has undergone extensive testing in pLGG. Initial trials established its optimal dosage and demonstrated encouraging outcomes in terms of partial responses and progression-free survival [33,34]. These results have led to additional studies, with emerging evidence suggesting the potential efficacy of MEK inhibitors even in the absence of characteristic BRAF mutations. Consequently, two major studies are currently evaluating selumetinib as a primary treatment option for pLGG. Other MEK inhibitors, such as trametinib, binimetinib, and cobimetinib, are also being explored for their applicability in pLGG [35,36]. While their deployment in treating low-grade gliomas is still in preliminary stages, the initial findings are promising. These inhibitors typically exhibit similar side effects, including dermatological and gastrointestinal reactions. Some, particularly in adult populations, have been associated with cardiac and ocular adverse effects [33,37]. The determination of the most effective MEK inhibitor for pLGG is still underway.

Direct BRAF inhibitors such as dabrafenib and vemurafenib also show potential for pLGG treatment. They specifically target BRAF kinases and have shown significant responses in pLGG with BRAFV600 mutations [38,39]. Ongoing studies are exploring these inhibitors for BRAF mutant pLGG. However, it is crucial to note that first-generation BRAF inhibitors might not be suitable for tumors with BRAF fusion due to potential adverse effects [40]. Second-generation inhibitors, which do not have this limitation, are being tested in ongoing trials and may offer a promising avenue [41]. Trametinib effectively treated progressive pLGG, achieving disease control in all subjects. Nonetheless, treatment-related side effects posed challenges for some patients, and a subset experienced disease recurrence after discontinuing MEKi [42] (Figure 2).

## 3. Specificities of WHO 2021 Classification of Brain Tumors

In the WHO CNS5 guidelines, the grading of central nervous system (CNS) tumors has been substantially revised: the transition from Roman to Arabic numerals for grading supersedes previous practices, and grading is now consistently implemented within specific tumor types rather than comparatively across different types. The significance of this specific change involves more clarity and universality for the classification, more precision and adaptability, and greater alignment with other classifications leading to their easier use in the research field, which ultimately benefits patients. This entity-specific grading approach for CNS tumors differs from other organ systems where neoplasms are graded according to type-specific systems, such as those for breast or prostate cancers [43]. The rationale behind adopting intra-type grading within WHO CNS5 is multifaceted: first, to provide greater grading flexibility relative to each tumor type; second, to emphasize the biological consistency within tumor types over the prediction of clinical behavior; and third, to synchronize with WHO’s grading protocols for non-CNS tumors [10]. In tandem with these grading modifications, nomenclature changes have been made to reflect molecular characteristics in accordance with cIMPACT-NOW Update 6 and to standardize terminology across all classifications within the WHO Blue Books, especially those pertaining to peripheral nerve and soft-tissue tumors [44].

The revised classification introduces fourteen new types within the categories of Gliomas, Glioneuronal Tumors, and Neuronal Tumors, along with updates to the nomenclature of existing entities. A key example is the reclassification of diffuse midline glioma, now termed “H3 K27-altered” instead of “H3 K27M-mutant,” to recognize a range of pathogenic mechanisms influencing these tumors [45].

Significantly, WHO CNS5 differentiates diffuse gliomas based on the patient’s age, distinguishing between “adult-type” and “pediatric-type”. This distinction acknowledges the clinical and molecular differences between these groups and aims to guide more effective treatment strategies for CNS tumors in both demographics [10]. Additionally, the classification now recognizes infant-type hemispheric glioma as a separate high-grade glioma category characterized by a unique molecular profile, including fusion genes involving ALK, ROS1, NTRK1/2/3, or MET, predominantly seen in newborns and infants [46].

## 4. Rare Entities in Low-Grade Gliomas

### 4.1. MYB/MYBL1 Alterations

Pediatric-type diffuse low-grade gliomas (pLGG) with MYB/MYBL1 alterations constitute a distinct subset of IDH-wild-type and H3-wild-type tumors, notable for their benign clinical course and favorable prognosis [47]. In 2021, the World Health Organization updated their CNS tumor classification to include two categories of these pLGGs: angiocentric glioma with MYB-QKI fusions and diffuse astrocytoma with various MYB/MYBL1 alterations [4]. Most of the existing studies on these gliomas have focused on their clinicopathologic characteristics, with less emphasis on their radiologic features [48]. The primary treatment strategy for pLGGs with MYB/MYBL1 alterations is comprehensive surgical resection, as complete removal is often correlated with a positive outcome [49].

The 2016 WHO update on CNS tumors offered valuable insights but did not thoroughly delineate pediatric gliomas and their prognostic outcomes. Specifically, the IDH-wild-type/H3-wild-type low-grade tumors remained a heterogeneous group. Despite their typically benign nature and rare progression to anaplastic forms in children, there was a lack of distinction between pediatric and adult tumor types. Research showed different molecular markers in tumors between children and adults, with pediatric low-grade gliomas predominantly exhibiting alterations in the BRAF, FGFR, and MYB/MYBL1 genes, while IDH1/2 mutations were less common [50]. This distinction was further emphasized by cIMPACT-NOW in their fourth update [47].

In its 2021 revision, the WHO introduced a classification for pediatric-type diffuse low-grade gliomas, encompassing four subtypes: (1) diffuse astrocytoma, MYB- or MYBL1-altered; (2) angiocentric glioma; (3) polymorphous low-grade neuroepithelial tumor of the young; and (4) diffuse low-grade, MAPK pathway-altered glioma [8]. This discussion focuses on the first subtype. There are few studies on the radiologic characteristics of MYB/MYBL1-altered gliomas. In a study by Chiang et al., 46 such tumors were evaluated, with 23 pre-operative MR images being reviewed. The majority of patients presented with epilepsy, and the tumors were predominantly located in the cerebral hemispheres, although some were found in the diencephalon and brainstem. Upon T1 imaging, these tumors typically appeared iso- to hypointense, while T2/FLAIR imaging often revealed mixed signals or hyperintensity. Only one case showed faint and diffuse contrast enhancement, and no diffusion restriction was observed [51]. In cases where complete resection is not possible, additional chemotherapy and radiation are considered. MYB/MYBL1 alterations can be considered distinctive in the field of oncology due to their unique molecular characteristics and implications, giving them an important role in the context of personalized medicine and hinting toward their potential as therapeutic targets.

### 4.2. Angiocentric Glioma

Angiocentric glioma (AG) is a unique brain tumor often associated with treatment-resistant epilepsy in children and young adults which can be effectively managed through neurosurgical intervention. An analysis of case reports since its initial identification revealed several key findings: (1) seizures are the most common initial symptom; (2) magnetic resonance imaging (MRI) typically reveals a supratentorial, non-enhancing lesion that is T1-hypointense and/or T2-hyperintense; (3) these tumors display specific histopathological features; and (4) outcomes following complete tumor resection are generally positive [52]. First identified in 2005 [4,53] and recognized as a distinct entity by 2007 [4], AG was initially categorized under “other glioma” in the 2016 WHO edition. However, in the latest classification, it is included among “pediatric-type low-grade diffuse gliomas”.

Due to the rarity of AG, gaining a comprehensive understanding has been challenging, but it is now graded as 1 in the 2021 WHO Classification. Commonly presenting with persistent, drug-resistant epilepsy in children, AG accounts for a small proportion of tumors in the German Neuropathology Reference Center [54]. A study by Kurokawa et al. reported a median patient age of 13. AGs are typically located in the supratentorial cortex, with a slight preference for the temporal lobe, although occurrences in the brainstem have been documented. MRI scans often reveal a single, T2-hyperintense lesion with no enhancement and a distinctive cortical rim on T1-weighted images [54,55].

Histologically, AG is characterized by an infiltrative growth pattern with uniform, bipolar spindle-shaped cells. Its hallmark features include perivascular cell arrangement around blood vessels and a horizontal cell stream beneath the pia-arachnoid structures. While some regions may resemble schwannomas, others can exhibit an epithelioid appearance. Key characteristics include the near absence of mitoses, microvascular proliferation, and necrosis. The tumor cells typically test positive for GFAP and negative for Olig2. EMA tests indicate ependymoma-like differentiation, corroborated by electron microscopy findings [53].

Some researchers postulate that AG originates from bipolar radial glia during embryogenesis, displaying ependymal features. Tests for IDH1-R132H, BRAF V600E, and neuronal antigens generally yield negative results, and the Ki-67 proliferation index is usually low. While rare anaplastic features have been noted, their clinical significance is not fully understood. Most AGs are associated with an MYB, QKI gene fusion, but the 2021 WHO Classification considers this only as a recommended, not mandatory, diagnostic criterion [4].

### 4.3. Diffuse Low-Grade MAPK Pathway-Altered Gliomas

The mitogen-activated protein kinase (MAPK) pathway is crucial in regulating a variety of cellular functions, including cell growth, differentiation, apoptosis, and more. This pathway is activated by signaling molecules such as FGF, EGF, IGF, and TGF binding to their respective cell surface receptors, initiating a cascade of cytoplasmic protein kinase activations. This series of activations leads to the phosphorylation of multiple proteins and nuclear transcription factors, ultimately affecting gene expression [56,57].

The dysregulation of the MAPK signaling pathway has been implicated in a range of diseases, including inflammatory, immunological, and degenerative disorders. Its aberration is also associated with the initiation and progression of various neoplasms due to factors such as abnormal receptor expression or genetic mutations activating receptors and downstream signaling molecules. This includes CNS tumors like pilocytic astrocytomas and gangliogliomas [58].

The recent WHO classification of CNS tumors has introduced a new category within pediatric-type diffuse low-grade gliomas: diffuse low-grade gliomas with MAPK pathway alterations. These tumors typically develop in childhood and can occur anywhere in the CNS, often presenting with epilepsy [59].

The exact prevalence of these tumors is somewhat uncertain, as specialized molecular testing is required for diagnosis, but they are considered relatively rare. Radiologically, they often appear as variably enhancing masses with cystic components. Histologically, these tumors exhibit diverse morphologies, usually displaying non-extensive infiltration patterns. On a molecular level, they are characterized by alterations in the genes associated with the MAPK pathway and are distinct in that they lack IDH1/2 and H3F3A mutations and CDKN2A deletion. Several subtypes of these tumors have been identified, with the most common alterations involving FGFR1 and BRAF mutations [9].

### 4.4. Polymorphous Low-Grade Neuroepithelial Tumor of the Young (PLNTY)

Polymorphous low-grade neuroepithelial tumor of the young (PLNTY) is an exceptionally rare, slowly progressing tumor that was recently incorporated into the World Health Organization classification of central nervous system tumors. Initially identified and characterized by Huse et al. in 2017, PLNTY was subsequently classified in the WHO Central Nervous System Tumors later that same year [60]. This tumor predominantly affects the temporal lobe (observed in approximately 80% of cases), although instances in other brain regions, like the parietal, frontal, and occipital lobes, have been documented. PLNTY typically presents in children and young adults, with an average age of onset around 20.6 years and a slight female predominance. It is categorized among long-term epilepsy-associated brain tumors (LEATs), which are commonly associated with seizures and often resistant to standard antiepileptic drugs [61]. However, symptoms of PLNTY may include headaches, dizziness, or visual disturbances.

Genetically, PLNTY is characterized by a unique DNA methylation profile and frequently involves alterations in the mitogen-activated protein kinase (MAPK) pathway, including the BRAF proto-oncogene and fibroblast growth factor receptors 2 and 3 (FGFR2 and FGFR3). These genetic alterations, such as BRAF-V600E mutations or FGFR2 and FGFR3 fusions, often coexist. BRAF-V600E mutations are more common in young adults, while FGFR2 fusions tend to be more prevalent in younger patients. The exact role of these genetic changes in the development of PLNTY is not fully understood [62,63].

The histology of PLNTY can vary, but it typically includes an oligodendroglioma-like component. This tumor type exhibits a range of cellular morphologies, from cells with uniformly small round nuclei to those with anisonucleosis or distinct nuclear features. Other features often observed include perivascular pseudorosetting and calcifications, while mitosis, necrosis, vascular proliferation, inflammation, and certain other cell features are typically absent. Immunostaining has shown positive staining for glial markers such as GFAP and Olig2, albeit with weak or focal expression, but CD34 expression was notably prominent and consistently observed across tumor cells and neuronal elements. Some tumor cells may exhibit antibodies for the BRAF p.V600E mutation, while the Ki-67 proliferation index is generally low, though higher values have been reported. Neuronal markers EMA and IDHp.R132H tend to be negative, and ATRX mutations and chromosome 1p/19q codeletion are absent as well [64].

## 5. Pediatric Low-Grade Gliomas: A Special Consideration

Tumors originating in the central nervous system (CNS) are the most commonly diagnosed solid tumors among children, with an estimated incidence rate of 5.4–5.6 cases per 100,000 individuals. These tumors can sometimes represent a cause of cancer-related mortality in this age group, with approximately 1 in every 100,000 diagnoses resulting in a fatal outcome. Among CNS tumors, pediatric-type low-grade gliomas (pLGGs) represent about 30% of brain tumor diagnoses in children. These tumors, classified as WHO grade 1 or 2 malignancies, encompass a variety of histological subtypes and can develop anywhere along the neural axis [65].

Children with low-grade gliomas typically present with both generalized and localized symptoms, often experiencing these symptoms for at least six months before diagnosis. General symptoms related to increased intracranial pressure due to ventricular obstruction include morning headaches, nausea, vomiting, and lethargy. Physical examination might reveal signs like impaired upward gaze, abnormalities of the sixth cranial nerve, or papilledema, often indicating tumor growth in regions such as the cerebellum, optic chiasm/hypothalamus, dorsally exophytic brainstem, or tectum. The manifestation of individual tumors varies depending on their location, frequently resulting in neurological deficits, seizures, and endocrinopathies in localized areas. For instance, cerebellar tumors often lead to ataxia and dysmetria, while cerebral hemisphere tumors may cause seizures, hemiparesis, or behavioral changes. Tumors affecting the hypothalamus and pituitary gland can lead to obesity, growth failure, diabetes insipidus, hormonal irregularities, and visual field impairment due to optic chiasm compression. Optic pathway gliomas, which can occur anywhere along the visual pathway, are more commonly bilateral or affect the chiasm and postchiasmatic regions in children with neurofibromatosis type 1. Symptoms of optic pathway gliomas include visual field impairments, reduced visual acuity, optic nerve atrophy, proptosis, or strabismus [66,67].

Brainstem low-grade gliomas typically progress slowly, often being detected after months to years. Although they do not extensively infiltrate the brainstem, dorsally exophytic and cervicomedullary tumors can cause lower cranial nerve deficits (e.g., dysphagia, dysarthria, abnormal breathing), as well as long tract signs such as hemiparesis, spasticity, hyperreflexia, and Babinski’s sign. Cervicomedullary tumors may also present with torticollis, long tract signs, and sensory loss due to upper cervical cord involvement; hydrocephalus is a common manifestation of focal brainstem tumors [68].

Upon neuroimaging, pediatric low-grade gliomas typically exhibit certain characteristics. MRI usually reveals these tumors to be hypointense on T1-weighted and hyperintense on T2-weighted sequences, with varying degrees of enhancement post-gadolinium. Pilocytic astrocytomas often appear as well-circumscribed tumors with cystic components and an enhancing nodule, while diffuse fibrillary astrocytomas are less well-defined and show lesser enhancement post-gadolinium. Accurate histological verification usually requires a surgical biopsy or complete tumor resection. In cases like optic pathway or hypothalamic gliomas in children, diagnostic biopsies might be avoided if MRI characteristics are consistent with low-grade glioma, particularly in the presence of neurofibromatosis type 1. Deep midline and brainstem tumor biopsies should be approached cautiously as these tumors often show no progression upon serial MRI evaluations [69].

Postoperative staging can sometimes involve an MRI scan of the surgical site within 24–48 h after surgery to differentiate between residual tumor and postoperative changes. In cases where dissemination or leptomeningeal involvement is suspected, a comprehensive evaluation should include spinal imaging and cerebrospinal fluid cytology testing [68]. A key feature of pediatric low-grade glioma (pLGG) is the abnormal activation of the mitogen-activated protein kinase (MAPK) pathway, suggesting that targeting this pathway with small-molecule inhibitors like MEK inhibitors could be a promising treatment strategy [42].

## 6. Treatment Modalities, Approaches, Outcomes, and Prognosis in Low-Grade Glioma

Achieving an optimal integrated diagnosis in neuro-oncology involves harmonizing histological categorization with genomic characterization. This process draws upon both histologically and genetically defined compendia of neoplasms. Despite the extensive nature of these compendia, certain correlations are commonly observed, with frequent integrations appearing in a manageable number of routine diagnoses. This approach is exemplified by the classification of ‘Diffuse low-grade glioma, MAPK pathway-altered’ as a specific tumor subtype [47].

In recent years, methylome profiling has emerged as a key method in CNS tumor classification. This technique, which analyzes genome-wide DNA methylation patterns, has gained significant attention in the academic field and is increasingly fundamental in the molecular taxonomy of CNS neoplasms [70]. While methylome profiling can sometimes serve as an indicator of genetic aberrations—for instance, a methylation signature akin to an IDH-wild-type glioblastoma may be identified without direct IDH mutation assays—it cannot completely replace mutation detection, especially in situations where targeted treatments or clinical trials require precise molecular aberrations [71]. Consequently, the molecular analysis of WHO grade II or III diffuse astrocytic, IDH-wild-type gliomas in adult patients is highly recommended. The presence of chromosomal aberrations such as +7/−10, EGFR amplification, or TERT promoter mutation should lead to a reclassification to WHO grade IV, significantly impacting both treatment strategies and prognostic expectations [72].

In pediatric low-grade glioma (pLGG), negative prognostic indicators include older age, astrocytic histology, large tumor size (>4–6 cm), midline crossing tumors, neurological deficits, and poor performance status. Conversely, presenting with seizures, particularly in neurologically intact individuals, is often viewed as a favorable prognostic factor. Pignatti et al. developed a scoring system in 2002, assigning points to various risk factors, and this system was validated across multiple trials [73]. The University of California, San Francisco’s (UCSF) more recent scoring system considers age, performance score, tumor size, and eloquent involvement in determining prognosis. Patients aged 55–60 years have a 5-year survival rate of 30% to 40%, with each additional year of age further diminishing their prognosis; however, those surviving beyond two years post-diagnosis may experience prolonged progression-free survival (PFS) despite challenging prognoses [74,75].

Tissue acquisition is crucial in accurately diagnosing, prognosing, and treating pLGG, as pathognomonic imaging is lacking. Needle biopsies can result in misdiagnosis rates of over 50%, making surgical resection the preferred method for tumor characterization. The support for extensive surgical resection is growing, as is evidence of its efficacy, although randomized controlled trials are still needed. This strategy was first proposed in 2001, and subsequent institutional studies, including one from the UCSF, have affirmed its effectiveness. Notably, the UCSF’s study demonstrated that a extent of resection (EOR) greater than 90% significantly improves overall survival (OS), with a 5-year survival rate of 97% versus 76% for EORs less than 90% [76]. The Johns Hopkins Hospital reported similar findings, indicating that gross total resection (GTR) can enhance both overall survival and progression-free survival (PFS). However, factors such as the involvement of the corticospinal tract, tumor volume, and oligodendroglioma histology can impede complete resection [77].

In a cohort study examining low-grade gliomas (LGGs), a significant correlation was found between both the residual volume post-surgery (*p* = 0.006) and the extent of surgical resection (*p* < 0.001) with overall survival among various LGGs. However, this correlation varied across the three LGG molecular subtypes. In the IDHmut-Codel subgroup, overall survival was significantly associated with the extent of resection (*p* = 0.01), but neither pre- nor postoperative tumor volumes showed a significant relationship. In contrast, in the IDHmut-Noncodel subgroup, preoperative volume (*p* = 0.018), postoperative volume (*p* = 0.004), and the degree of resection (*p* = 0.002) each were associated with overall survival. For the IDHwt subtype, there was no significant association between tumor volumes or resection extent and overall survival [78].

The relationship between the extent of surgical resection and overall survival is particularly noted in molecularly characterized IDH mutant astrocytomas and oligodendrogliomas. This association appears more pronounced in astrocytomas, potentially because of the higher efficacy of non-surgical therapies in oligodendrogliomas or their generally longer survival periods, which could mask the survival benefits of surgical intervention [79,80]. Patel et al. reported in their 2018 study involving a cohort of 74 patients with WHO grade II diffuse gliomas that the extent of glioma resection correlated with overall survival in the IDH-wild-type subgroup but not in the IDH mutant subgroup. However, this study had limitations, such as an incomplete description of IDH mutation testing protocols and a lack of stratification by 1p/19q-codeletion status [81].

Prospective trials and retrospective studies have not consistently shown the significant prognostic effects of extent of resection (EOR) on overall survival (OS) and progression-free survival (PFS), but cognitive and quality of life outcomes post-surgery remain important considerations. The average preoperative cognitive function score in the LGG cohort, as measured by the EORTC score, was 80.9, compared to 70.9 in the high-grade glioma (HGG) group. Postoperatively, the LGG group’s scores remained stable, while the HGG group showed significant improvement at 1- and 6-month follow-ups. In the LGG cohort, cognitive function changes varied, with 24% reporting improvement and 20% experiencing deterioration at 1 month postoperatively [82]. The rapid growth rate of IDH-wild-type gliomas may exert more pressure on adjacent brain structures than IDH mutant gliomas, suggesting that more aggressive surgical resection could improve cognitive outcomes by relieving mass effects and associated edema [83]. Postoperative experiences differ among patients, with some experiencing relief and others facing the stress of cancer diagnosis and ongoing surveillance or treatment. Notably, lower preoperative cognitive function scores have been observed in females compared to males [84].

Neuronavigation and brain mapping technologies, including functional MRI and cortical stimulation mapping, aid in precise resections while preserving quality of life. Neurosurgeons can customize procedures to individual brain structures, thereby minimizing permanent deficits. Brain mapping has shown efficacy in reducing permanent deficit rates, increasing gross total resection (GTR) rates, and providing survival benefits. Ideally, a prospective, multicenter trial would address this issue definitively, but challenges in recruitment, follow-up, and ethical considerations make organizing such a trial complex [85,86].

In neuro-oncology, temozolomide has gained attention as a chemotherapy drug, especially due to its ease of oral administration, lower toxicity compared to PCV (procarbazine, lomustine, and vincristine), effective penetration of the blood–brain barrier, and proven effectiveness against glioblastoma. Phase 2 studies have shown temozolomide to be effective against growing LGGs, whether previously exposed to radiation or not, on standard 5-day or alternate schedules like 3 weeks on followed by 1 week off, or 7 weeks on followed by 4 weeks off. Temozolomide has also been associated with improved quality of life outcomes [87].

In the realm of glioma treatment, there exist pivotal inquiries concerning the potential of temozolomide to either supplant radiotherapy or complement it in the management of low-grade gliomas (LGGs). Presently, ongoing clinical trials are diligently endeavoring to elucidate these quandaries. A phase 3 investigation spearheaded by a consortium of European and Canadian researchers is actively scrutinizing this matter by juxtaposing radiotherapy against temozolomide therapy for individuals afflicted with LGGs, with careful consideration being given to the chromosomal 1p status. This comprehensive study aims to assess a gamut of clinical outcomes, encompassing the likes of progression-free survival (PFS), neurocognitive functionality, and overall quality of life [88,89]. Furthermore, there are concerted endeavors to ascertain the advantages of amalgamating temozolomide with radiotherapy, particularly in the context of high-risk LGGs. The Radiation Therapy Oncology Group (RTOG) has successfully concluded its phase 2 inquiry (RTOG 0424), while the Eastern Cooperative Oncology Group (ECOG) has embarked upon a phase 3 exploration (ECOG E3F05). The overarching objective of these initiatives, in conjunction with similar studies unfolding in Europe, is to elucidate the role that temozolomide plays within the treatment paradigm for LGG [88,89].

In a separate investigation, a phase II trial delineated its primary objective as evaluating the response to temozolomide (TMZ) among pediatric patients grappling with recurrent or progressive LGG. The inception of this trial emanated from the Preston Robert Tisch Brain Tumor Center at Duke University Medical Center and subsequently expanded to encompass additional clinical sites. Notably, TMZ was administered orally under fasting conditions, with treatment cycles recurring at 28-day intervals. The observed outcomes encompassed partial response (PR) in three patients and minimal response (MR) in one patient, while 42% of patients exhibited stable disease (SD), and an equivalent percentage showed progressive disease (PD) after a minimum of two treatment cycles [90].

In a tangentially related vein, there exists substantiating evidence derived from the RTOG trial (RTOG 9802) which underscores the potential of employing procarbazine, lomustine, and vincristine (PCV) in tandem with radiotherapy, particularly in the context of recurrent LGGs post-radiotherapy. In this investigation, individuals who received a combined regimen of PCV and radiotherapy exhibited more favorable outcomes in terms of progression-free survival (PFS). Nevertheless, there was no statistically significant disparity in overall survival, thereby suggesting that PCV may serve as a potent adjunct both as a secondary intervention and when administered concomitantly with radiotherapy. It is imperative to note, however, that there exists a dearth of consensus regarding the optimal timing of surgery and its overarching impact on LGG management, necessitating further comprehensive exploration through prospective studies, mirroring the scrutiny accorded to the timing of radiotherapy in the treatment of LGGs [91].

It is paramount to acknowledge that radiotherapy stands as the sole therapeutic modality validated through a randomized controlled trial to confer certain advantages upon patients grappling with LGGs. Nonetheless, the optimal utilization of radiotherapy remains a topic of incessant deliberation. The EORTC 22845 study has proffered insights into this discourse, demonstrating that individuals subjected to early radiotherapy (54 Gy) experienced prolonged intervals devoid of disease progression (PFS) and exhibited superior seizure control relative to those subjected to delayed radiotherapy. Concretely, the progression-free survival stretched to 5.3 years for the early treatment cohort as opposed to 3.4 years for their delayed treatment counterparts (*p*  <  0.0001). Furthermore, a noteworthy 75% of individuals in the early treatment cohort achieved seizure control in comparison to 59% in the delayed treatment cohort (*p*  =  0.0329). Despite these discernible benefits, there was no marked discrepancy in overall survival between the two cohorts, with values of 7.4 years for the early cohort and 7.2 years for the delayed cohort. Given the absence of definitive data regarding quality of life, researchers have proffered the contention that it may be reasonable to defer radiotherapy for LGG patients who are in robust health. This hesitation emanates from the ambiguous equilibrium between the advantages inherent to extended progression-free survival and seizure control and the potential merits associated with overall survival. Additionally, it is worth noting that 35% of patients slated for deferred radiotherapy ultimately circumvented its necessity, thereby mitigating potential side effects [92].

Recent studies have cast a focused spotlight upon the evaluation of quality of life post-radiotherapy to gain deeper insights into its ramifications for individuals afflicted by LGGs. A phenomenon known as radiation leukoencephalopathy, which may manifest months or even years subsequent to cranial radiotherapy, is typified by a gradual decline in multifarious domains, including personality, equilibrium, urinary continence, attention, memory, and higher-order cognitive faculties [93]. To ameliorate these deleterious sequelae, select studies proffer the notion that through meticulous adjustments of total dosage, sessional dose, and irradiation field, it is feasible to uphold treatment efficacy while concurrently attenuating associated risks [94]. Nevertheless, in light of the relatively protracted overall survival (OS) rates observed among LGG patients, the potential of encountering these complications remains palpable. A recent comprehensive inquiry conducted by Douw et al. [95] undertook an exhaustive analysis of cognitive and quality of life outcomes among 65 LGG patients, with half having undergone radiotherapy. Over an average observation period spanning 12 years, the study unearthed that 27% of non-irradiated patients manifested substantive cognitive impairments in at least 5 of the 18 evaluated parameters. In stark contrast, this proportion burgeoned to 53% for those who had received radiotherapy. Predominant deficits were observed in the realms of cognitive processing and attention, with other noticeable, albeit statistically non-significant, declines detected in information processing speed, motor dexterity, and working memory [91,95].

In the sphere of pediatric neuro-oncology, the emergence of molecularly targeted treatments tailored for pediatric low-grade gliomas (pLGGs) has been greeted with considerable enthusiasm. These therapeutic interventions, with a specific focus on the dysregulated Ras-MAPK pathway, exemplified by RAF inhibitors and MEK inhibitors, are either receiving validation from the FDA or undergoing rigorous clinical evaluations for their applicability in the context of pLGGs [96,97]. However, it is of paramount significance to underscore that first-generation Type 1 BRAF inhibitors are not recommended for pLGGs characterized by BRAF rearrangements due to their proclivity to incite the paradoxical activation of the MAPK pathway via heightened RAF dimerization [98].

The PNOC001 phase II study, which embarked upon an investigation into the efficacy of the mTOR pathway inhibitor everolimus in cases of recurrent or progressive pLGG, charted pioneering territory by mandating a prerequisite for tissue diagnosis [99]. Subsequently, PNOC014 emerged as the inaugural trial tasked with scrutinizing the safety profile of a Pan-RAF inhibitor among pediatric patients grappling with LGG. The auspicious findings gleaned from the initial cohort of patients have expedited the progression to PNOC026/Day101-001—a phase II study singularly dedicated to appraising the oral Pan-RAF inhibitor (Day101) in individuals afflicted by recurrent or progressive pLGGs characterized by BRAF alterations [100]. Furthermore, therapeutic agents designed specifically to target the BRAF V600E mutation, such as dabrafenib and vemurafenib, have demonstrated encouraging outcomes in early-phase clinical trials involving patients with pLGGs. A recent revelation stemming from the phase II trial presented by Bouffet et al. at the American Society of Clinical Oncology (ASCO) Annual Meeting unveiled a noteworthy overall response rate (ORR) for the combination therapy of dabrafenib and trametinib (47%), signifying a substantial enhancement in comparison to the ORR associated with the conventional chemotherapy regimen employing carboplatin and vincristine (11%) [101].

It merits mention that therapeutic agents custom-tailored to target aberrant cellular pathways in pediatric low-grade gliomas (pLGGs) exhibit a toxicity spectrum that diverges markedly from that encountered with traditional chemotherapeutic regimens. Traditional chemotherapy regimens for pLGGs, while efficacious, are often accompanied by a constellation of adverse effects, encompassing myelosuppression, alopecia, ototoxicity—particularly notable with the utilization of carboplatin—and, although less frequently observed, perturbations in fertility potential, notably associated with procarbazine [102]. Conversely, targeted therapeutic modalities such as MEK and BRAF inhibitors give rise to a distinct set of side effects, which encompass dermatological toxicities, elevations in creatine phosphokinase (CPK), cardiovascular complications, and ocular adverse events [103].

## 7. Conclusions

The landscape of LGG treatment is undergoing a transformative shift. Emerging strategies challenge traditional methods, questioning the risks of a less dynamic approach and the direct implications of radiotherapy while highlighting the merits of proactive measures like comprehensive surgical removal and initial chemotherapy. Given the current data, a compelling approach might be to prioritize extensive surgery when feasible and reserve radiotherapy for the point of disease advancement. Ongoing clinical trials hold the promise of redefining LGG treatment, particularly spotlighting the potential role of temozolomide, which might even negate the necessity for radiotherapy in the future. It is imperative that future research delves deeper, leveraging advanced imaging and molecular markers to decode prognoses more accurately.

## Figures and Tables

**Figure 1 brainsci-13-01700-f001:**
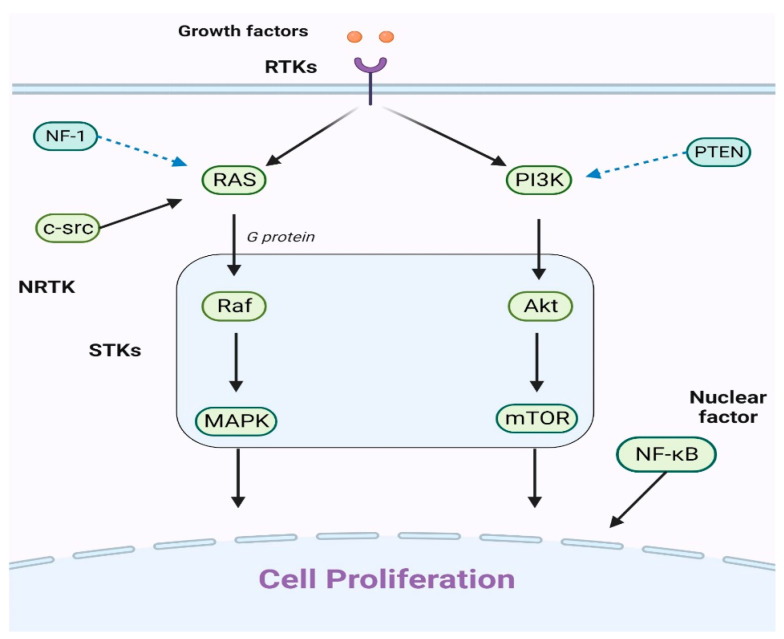
An explanation of the primary cell proliferation pathways: The PI3K/Akt/mTOR and Ras/Raf/MAPK routes are the main pathways. When growth factors attach to receptor tyrosine kinases (RTKs), they can trigger either the Ras/Raf/MAPK or PI3K/Akt/mTOR pathways. The key players in these pathways, Raf, MAPK, Akt, and mTOR, have been identified as serine/threonine-specific protein kinases (STKs). Additionally, the intracellular tyrosine kinase c-src can initiate the Ras/Raf/MAPK pathway. It is worht noting that the nuclear factor NF-κB also significantly contributes to cell proliferation.

**Figure 2 brainsci-13-01700-f002:**
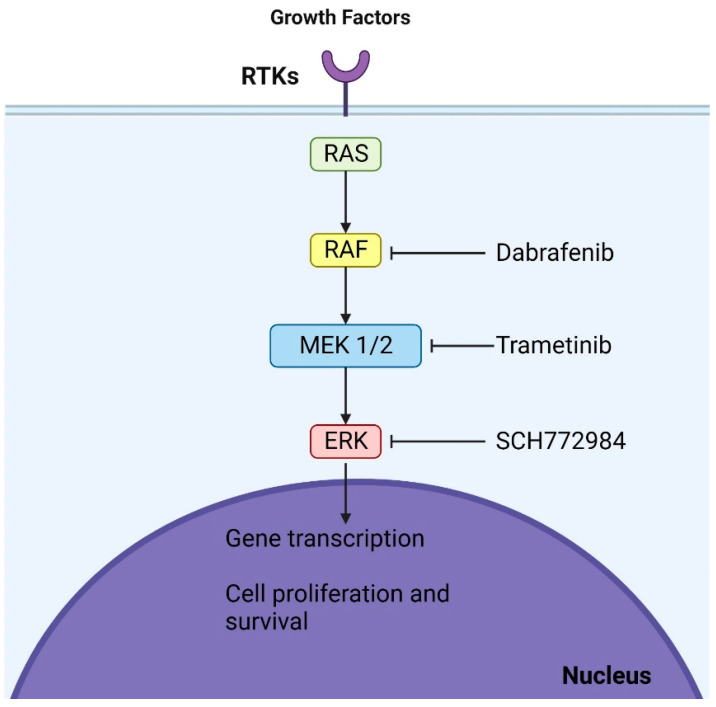
Mechanisms of action of dabrafenib and trametinib: These agents, which are BRAF and MEK inhibitors, respectively, act at two distinct sites within the MAPK (mitogen-activated protein kinase) pathway. By binding to their respective targets, they halt the oncogenic signaling cascade, culminating in cell cycle arrest.

## Data Availability

All data are available online on libraries such as PubMed.

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
