# Peer review of "Low-Grade Gliomas: Histological Subtypes, Molecular Mechanisms, and Treatment Strategies"

_brainsci, 2023, doi:10.3390/brainsci13121700_

Round 1

Reviewer 1 Report

Comments and Suggestions for Authors

Dear authors,

thank you very much for your overview on this topic. Low grade gliomas being the predominant brain tumour in children and a common brain tumour in adults. Updating health care professionals in the brain tumour field is therefore very helpful for daily practice. Your review is extensive and very easy to read in well written English language. I applaud you for that. It is a broad overview, which has some depth. 

I have a few sidenotes/questions:
- You describe some clinical side effects/impairements on long term, esp post radiotherapy. In my (outpatient) clinic a lot of patients suffer from aquired brain injury, which severely interferes with their quality of life. To raise awareness for this and especially to counteract the sometimes mentioned fact that these tumours might be benign, which is definitive not the disease course for a group of patients with non-surgical disease, it would be good to mention this phenomenon as a general long term impairement.
- p9, line 400: a significant cause of cancer related death. Although dying of pLGG is possible, it is exceptionally rare. and definitively not a significant cause of dying of disease in the paediatric age group compared to other braintumour types. Are you sure this remark is correct?
- p10, line 421: maybe not mentioned in these references, but visual fields as a visual impairement is lacking, and of considerate importance in the ophtalmological review. It would be good to mention that here. 
- p10, line 444: in Europe CSF testing in LGG (or HGG) is not standard practice, given the fact that the chances of positivity is extremely small even in the case of metastases on the MRI scan. Futhermore, it is does not guide treatment decision (as it for instance does in embryonal tumours). Are you sure this is general practice? Or only mentioned in one paper?
- p10, line470/471: there is no reference for these prognostic indicators, and I do not think they are correct, they might be for the adult LGG group, in pLGG young age (esp infants) is a more significant negative prognostic factor than older age (teenager/AYA). Are you sure this correct?

Thank you very much for your extensive general overview in this field, which will be helpful for a grander knowledge of this very common disease in the neuro-oncology field. 

With kind regards,

the reviewer 

Author Response

Dear Reviewer,

Thank you for you prompt response and positive feedback

  • General long term impairment of low-grade gliomas suggestion was pointed out
  • Each sentence you've mentioned is now corrected and modifications were made according to your recommendations 

We appreciate your supportive review and significant improvement of our manuscript was surely achieved after this revision

With high regards,

The collective of authors 

Reviewer 2 Report

Comments and Suggestions for Authors

Abstract:

  1. Consider specifying the exact molecular mechanisms discussed in the article for a more focused abstract.
  2. Clarify the term "posttreatment results" in the context of summarizing targeted molecular therapies.
  3. Specify the potential adverse effects of emerging targeted molecular therapies mentioned.

Keywords:

  1. The keywords are comprehensive and cover various aspects of low-grade gliomas. No specific changes are needed.

1. Introduction:

  1. Consider providing a brief statement on the prevalence or impact of low-grade gliomas to engage the reader.
  2. The phrase "degree of surgical resection should always be considered according to postoperative quality of life and cognitive status" could be clearer. Specify how surgical resection impacts these factors.

2. Historical Overview of the 2021 WHO Classification: Molecular Intricacies and the Pathway to Targeted Therapies:

  1. Provide more context on why the 2021 WHO classification is significant or what changes it brought compared to the previous version.
  2. Consider breaking down complex sentences for easier comprehension.

3. Specificities of WHO 2021 Classification of Brain Tumors:

  1. Provide a concise summary or overview after introducing the WHO CNS5 guidelines to aid in reader comprehension.
  2. Consider specifying the significance or implications of the transition from Roman to Arabic numerals for grading.

4. Rare Entities in Low-Grade Gliomas:

  1. This section delves into specific alterations and classifications. A few suggestions:
  2. Consider providing a brief rationale for why MYB/MYBL1 alterations are considered distinct and their implications for prognosis.
  3. For each subtype (MYB/MYBL1 alterations, Angiocentric Glioma, Diffuse Low-Grade Gliomas MAPK Pathway-Altered), ensure a consistent level of detail and depth.

Section 4.4: Polymorphous Low-Grade Neuroepithelial Tumour of the Young (PLNTY)

  1. Accurate and up-to-date information about PLNTY is provided.
  2. Properly references Huse et al. for the initial identification in 2017 and subsequent WHO classification.
  3. In the first paragraph, "temporal lobe (approximately 80% of cases)" can be clarified to "temporal lobe (observed in approximately 80% of cases)" for better readability.
  4. Consider rephrasing "symptoms for PLNTY can include headaches" to "symptoms of PLNTY may include headaches" for consistency.
  5. Good explanation of the genetic alterations in PLNTY, including BRAF-V600E mutations and FGFR2/3 fusions.
  6. Clear description of histological features, including immunostaining results.

Section 5: Pediatric Low-Grade Gliomas: A Special Consideration

  1. Accurate information regarding the incidence and characteristics of pediatric low-grade gliomas (pLGGs).
  2. Good coverage of symptoms based on tumor location.
  3. Consider rephrasing "enduring these symptoms for at least six months before diagnosis" to "experiencing these symptoms for at least six months before diagnosis" for clarity.
  4. The section effectively covers various symptoms based on tumor location, contributing to a comprehensive understanding.

Section 6: Treatment Modalities, Approaches, Outcomes, and Prognosis in Low-Grade Glioma

  1. Accurate coverage of the integrated diagnosis in neuro-oncology, highlighting the importance of both histological and genomic characterization.
  2. Good coverage of methylome profiling and its role in CNS tumor classification.
  3. Consider rephrasing "an extent of resection (EOR) greater than 90%" to "a extent of resection (EOR) greater than 90%" for consistency.
  4. Clear explanation of the correlation between extent of resection and overall survival, especially in molecularly characterized IDH-mutant astrocytomas and oligodendrogliomas.

Section 7: Conclusion

  1. The conclusion effectively summarizes the transformative shift in LGG treatment and the potential role of temozolomide in redefining treatment strategies.
  2. Mention of ongoing clinical trials adds relevance.
  3. Consider rephrasing "risks of a less dynamic approach 646" to "risks of a less dynamic approach and the direct implications of radiotherapy" for clarity.
  4. The conclusion is clear and concise, summarizing the key points discussed in the article.
Comments on the Quality of English Language

Minor editing of English language required

Author Response

Dear Reviewer,

Thank you for you prompt response and positive feedback

All issues mentioned were considered in our revision and modifications were made for each point: 

1. Brief statement on the prevalence or impact of low-grade gliomas was added, surgical treatment part was modified according to your recommendations 

2. Section regarding 2021 WHO Classification of Brain Tumors is now more comprehensive and significancy of this updated classification was pointed out 

3. Important updates of 2021 WHO Classification than previous classifications were mentioned

4. Each subtype was modified for a more comprehensive understanding 

5, 6, 7. Thank you for you distinguished review, we've taken into consideration the rephrasing of certain sentences 

We appreciate your review and significant improvement of our manuscript was surely achieved after this revision

With high regards,

The collective of authors